# Clinical Relevance of Trace-Positive Results in Xpert MTB/RIF Ultra for Tuberculosis Diagnosis in a High-Burden Setting: A Retrospective Cohort Study

**DOI:** 10.3390/diagnostics15222860

**Published:** 2025-11-12

**Authors:** Cristian Sava, Alin Iuhas, Cristian Marinău, Radu Galiș, Marius Rus, Mihaela Sava

**Affiliations:** 1Department of Medical Disciplines, Faculty of Medicine and Pharmacy, University of Oradea, 410073 Oradea, Romania; 2Bihor County Clinical Emergency Hospital, 410167 Oradea, Romania

**Keywords:** tuberculosis, Xpert MTB/RIF Ultra, trace call, high-burden settings

## Abstract

**Background**: The introduction of the “trace” category in the Xpert MTB/RIF Ultra assay has significantly improved the sensitivity of molecular tuberculosis diagnostics. While it enhances sensitivity, especially in paucibacillary and extrapulmonary cases, its specificity remains debatable, making its interpretation outside select populations a topic of clinical uncertainty. **Objectives**: This study evaluates the diagnostic and clinical significance of trace-positive results obtained with the Xpert MTB/RIF Ultra assay in the context of a high-incidence TB setting, examining their association with clinical, imaging, and microbiological findings. **Methods**: A retrospective analysis was conducted on 65 samples with trace-positive Xpert Ultra results, collected over a six-year period from 59 distinct patients in a general hospital in Romania. Correlations were assessed with microscopy, culture, clinical features, imaging, treatment initiation, and prior TB history. A composite reference standard was used for diagnostic accuracy evaluation. **Results**: Of the 65 trace-positive samples, 29 (44.6%) were culture-positive and 5 (7.7%) were smear-positive. A high proportion of patients, 56 (94.9%), presented with TB-compatible symptoms, and 47 (79.6% of those with imaging) had highly suggestive radiological findings. Based on the composite reference standard, 47 patients (79.7%) were ultimately diagnosed with active TB. Anti-TB treatment was initiated in 44 patients (74.5%). Trace positivity was observed across various specimen types, including sputum, pleural fluid, and cerebrospinal fluid. **Conclusions**: In high TB burden environments, trace-positive Xpert Ultra results frequently reflect true disease when interpreted within the appropriate clinical and imaging framework. Our findings indicate that, in regions with high tuberculosis incidence such as Romania, trace-positive Xpert Ultra results may contribute meaningfully to clinical decision-making when interpreted alongside clinical and radiological findings, in alignment with current WHO guidance.

## 1. Introduction

Tuberculosis (TB) continues to pose a critical global health threat, remaining one of the top ten causes of death worldwide. According to the Global Tuberculosis Report 2024 by the World Health Organization (WHO), an estimated 10.8 million people developed TB in 2023, and 1.25 million died due to the disease. This reflects a global incidence rate of approximately 134 cases per 100,000 population, underscoring the persistent burden of tuberculosis as a major public health challenge despite ongoing efforts in prevention, diagnosis, and treatment. The burden remains disproportionately high in low- and middle-income countries, particularly in Southeast Asia and sub-Saharan Africa [1]. In 2022, all European Union and European Economic Area (EU/EEA) countries reported TB notification data resulting in a notification rate of 8.0 per 100,000 population. Romania, with an incidence rate of 55 per 100,000 population, remains among the highest in the European Union [2,3]. However, real-world data specifically evaluating the diagnostic performance and clinical interpretation of the highly sensitive Xpert Ultra trace call in this high-burden European context are scarce, highlighting a significant gap in regional evidence.

The early diagnosis of TB is critical for initiating appropriate treatment promptly, reducing morbidity and mortality, and curbing further transmission of the disease [4,5]. Given that TB remains one of the top infectious disease killers globally, timely identification of active cases is a cornerstone of TB control efforts. Traditional diagnostic methods, such as sputum smear microscopy and culture, while widely used, are often limited by low sensitivity in paucibacillary disease and long turnaround times, respectively [6,7]. Advances in molecular technology have significantly enhance the rapid diagnosis of infectious diseases, including TB [8,9]. Consequently, WHO now endorses several rapid molecular tests for clinical use. Among these, the automated nucleic acid amplification test Xpert MTB/RIF and its next-generation version, Xpert MTB/RIF Ultra (Cepheid, Sunnyvale, CA, USA), are recommended as first-line diagnostic tools for both pulmonary and extrapulmonary TB in adults and children [10]. These molecular platforms have substantially improved diagnostic accuracy, particularly for pulmonary TB and the detection of drug resistance [11].

The original Xpert MTB/RIF assay, launched in 2010, was a landmark in TB diagnostics, offering rapid molecular detection of *Mycobacterium tuberculosis* complex (MTBC) and rifampicin resistance in a single cartridge-based platform. Its clinical utility was rapidly recognized for sputum-positive pulmonary TB. However, its sensitivity was suboptimal in smear-negative, extrapulmonary, and pediatric TB—areas where diagnostic delays are most consequential [12,13]. In response, the Xpert MTB/RIF Ultra was developed with key improvements: targeting multi-copy DNA sequences (IS6110 and IS1081) instead of the single-copy rpoB gene alone; lower limit of detection (~15.6 CFU/mL compared to 114 CFU/mL with the original assay); addition of a “trace” category, indicating detection of very low levels of MTB DNA below the threshold for rifampicin resistance testing [14].

While the inclusion of a “trace” category in the Xpert MTB/RIF Ultra assay significantly expands diagnostic reach, it also introduces interpretative uncertainty. A trace result may indicate early or active TB, residual DNA from prior infection, or, in rare cases, a false-positive signal. The clinical relevance of such results depends strongly on the sample type, patient history, and pre-test probability of disease. Evidence suggests that trace-positive findings can carry significant diagnostic weight when interpreted within an appropriate clinical and radiological framework [15,16]. In children, trace-positive results from Xpert MTB/RIF Ultra are increasingly recognized as true-positive findings, given the low bacillary load and frequent absence of microbiological confirmation. This interpretation, supported by international guidelines and recent studies, highlights the diagnostic relevance of trace results when supported by clinical presentation and imaging suggestive of TB [17,18,19]. Similarly, among people living with HIV (PLHIV), the sensitivity of Xpert Ultra, including trace-level detection, is superior to that of both conventional Xpert and culture, particularly in case of sputum-scarce or disseminated disease [20,21]. Moreover, trace results have been associated with an increased diagnostic yield in extrapulmonary TB (EPTB), particularly when testing specimens with inherently low mycobacterial burden, such as cerebrospinal fluid, lymph node aspirates, and pleural fluid. Although sensitivity varies depending on the specimen type, disease site, and immune status of the patient, multiple studies have demonstrated that the detection of trace amounts of *Mycobacterium tuberculosis* DNA can provide critical diagnostic value. This is especially relevant when interpreted in conjunction with clinical symptoms, radiological findings, and epidemiological context, where microbiological confirmation is often difficult to obtain by conventional methods [16,18,22].

Beyond the high-risk groups previously discussed, the clinical interpretation of trace-positive results remains challenging. In immunocompetent adults without risk factors, particularly those with a history of TB, trace detection may reflect residual nonviable DNA rather than active disease, especially in the absence of symptoms or radiological findings [21,23]. Although the enhanced sensitivity of Xpert MTB/RIF Ultra improves case detection in paucibacillary forms, it comes at the cost of reduced specificity, particularly in previously treated individuals [24]. In high-burden settings, where subclinical presentations and undocumented prior TB are common, distinguishing early disease from false positivity becomes particularly difficult [25].

Despite these concerns, several studies have demonstrated that trace results, even outside WHO-prioritized groups, may correlate with active TB. In selected cases, particularly those with compatible clinical and radiological features, trace positivity has supported treatment initiation with favorable outcomes [23,26,27]. This underscores the importance of contextual interpretation, integrating microbiological, clinical, radiological, and epidemiological findings to guide management decisions [28]. This aligns with the evolving 2025 WHO guidelines, which place an increased weight on trace results when supported by the overall clinical picture, marking a shift from earlier, more conservative recommendations. Such integrated approaches move beyond standard algorithmic diagnostics, promoting a personalized medicine perspective essential for optimizing timely treatment in complex, real-world settings [29,30,31].

Considering the ongoing uncertainty regarding the interpretation of Xpert MTB/RIF Ultra trace results and their potential clinical significance even beyond WHO-prioritized groups, we conducted a retrospective observational study, carried out in a high TB prevalence setting. The objective of this study was to evaluate the correlation between these findings and clinical, radiological, and microbiological indicators of active TB, and to determine the circumstances under which such results can reliably guide TB diagnosis and treatment decisions. Specifically, we evaluated the proportion of trace-positive samples that were microbiologically confirmed by culture and/or smear microscopy, and examined correlations between trace results and TB-specific symptoms, imaging findings, and disease localization. We also explored associations with prior TB history, comorbidities, and the type of biological specimen analyzed. Furthermore, we assessed the frequency and context of treatment initiation following a trace result and investigated whether the use of a composite reference standard (CRS) could enhance diagnostic interpretation compared to conventional culture alone. By analyzing diverse cases from our institution, we provide real-world insights into how these results correlate with clinical, radiological, and microbiological indicators of active TB. The aim of this study was to evaluate the diagnostic and clinical significance of trace-positive Xpert Ultra results using a composite reference standard in a high-burden European setting.

## 2. Materials and Methods

### 2.1. Design

We conducted a retrospective, observational study between January 2019 and June 2025 in a general hospital in Romania, which includes a specialized TB diagnosis and treatment unit. The hospital serves both urban and rural populations and functions as a reference center for molecular TB diagnosis in the region.

### 2.2. Patient and Sample Selection

Patients were selected for inclusion if they met the following criteria:•They were evaluated as presumptive TB patients at the time of diagnostic workup.•At least one clinical specimen was tested using Xpert MTB/RIF Ultra and returned a “trace” result.•Comprehensive clinical records, imaging data, and microbiological workup results were available for retrospective review.

Exclusion Criteria: Patients were excluded if they were already receiving anti-TB treatment at the time the Xpert Ultra trace result was generated, or if clinical data were incomplete or unavailable.

The final cohort included a total of 65 trace-positive samples obtained from 59 distinct patients who met the inclusion criteria. Six patients contributed two samples each to the final analysis, collected either simultaneously from different sites or at different time points during diagnostic follow-up due to persistent or recurrent symptoms.

### 2.3. Diagnostic Methods

#### 2.3.1. Bacteriological Examination

All samples underwent:•Microscopy using Ziehl–Neelsen staining•Culture on Löwenstein–Jensen medium or liquid medium culture•Xpert MTB/RIF Ultra testing as per manufacturer protocol

Trace results were defined as amplification of IS6110/IS1081 with no detectable rpoB signal and high Ct values, without rifampicin resistance interpretation.

#### 2.3.2. Clinical and Radiological Evaluation

Patient records were reviewed for:•Respiratory and systemic symptoms (cough, dyspnea, sputum, hemoptysis, fever, night sweats, fatigue, weight loss)•Comorbidities (oncological, immunological, pulmonary, cardiovascular)•TB history (prior treatment, default, or relapse)•Radiological findings (chest X-ray or CT)—Images were interpreted by radiologists blinded to culture results and categorized as either suggestive or non-suggestive for TB.

#### 2.3.3. Composite Reference Standard (CRS)

Because mycobacterial culture is an imperfect gold standard limited by slow growth, loss of viability during transport, reduced sensitivity in paucibacillary and extrapulmonary TB, a predefined composite reference standard was applied to more accurately classify samples, integrating microbiological confirmation (culture or smear), clinical symptoms consistent with WHO TB definitions (fever, cough lasting longer than 2 weeks, weight loss, hemoptysis, night sweats, fatigue, dyspnea, chest pain), and radiological features (radiography, computed tomography) suggestive of TB (cavitary lesions, miliary patterns, infiltrates, pleural effusion, adenopathy, caseous nodules). The decision to initiate treatment, as documented in the hospital TB registry, was included as supportive evidence within the CRS classification.

Patients were classified as follows:•Confirmed TB: positive culture, both solid or liquid (smear positive/negative);•Probable TB: negative or unavailable culture but strong clinical and radiological evidence of active TB, as determined by an independent clinician;•Not TB: absence of microbiological, clinical, or radiological findings compatible with TB.

The CRS was developed in alignment with WHO diagnostic guidance [5] and previous methodological frameworks [5,32,33].

The index test (Xpert MTB/RIF Ultra trace results) was not included in the CRS to avoid incorporation bias. This approach allowed estimation of Ultra performance against a composite reference independent of the index test, reflecting real-world diagnostic decision-making in a high-burden setting [5,34].

#### 2.3.4. Ethics

The study was conducted in accordance with the Declaration of Helsinki, and approved by the ethics committee of Bihor County Clinical Emergency Hospital (protocol code 3974, approval date 6 February 2025). Informed consent was obtained from all subjects involved in the study. All data were processed and stored strictly in accordance with national and EU privacy standards (GDPR).

Declaration on the Use of Generative AI: The authors used generative artificial intelligence (ChatGPT, OpenAI’s GPT-5 model (ChatGPT Plus), OpenAI) exclusively for language editing, improving clarity, and rephrasing certain segments of the manuscript. The AI tool was not used to generate original scientific content, data, figures, or to assist in study design, data analysis, or interpretation of results. All scientific decisions, data interpretation, and manuscript conclusions were made solely by the authors.

## 3. Results

### 3.1. Study Cohort, Trace Incidence, and Specimen Characteristics

Over the study period (January 2019–June 2025), a total of 2683 Xpert Ultra tests were performed. Of these, 596 (22.2%) were positive (any call), with 65 samples yielding a “trace” result. The trace results accounted for 2.4% of all Xpert Ultra tests performed and 10.9% of all positive results. These 65 samples originated from 59 distinct patients and formed the study cohort.

The distribution of the trace-positive specimens is detailed in Table 1. The majority of samples were respiratory: spontaneous sputum (30, 46.2%) and induced sputum (23, 35.4%). Extrapulmonary specimens (12, 18.5%) included pleural fluid (6, 9.2%), bronchial aspirate (3, 4.6%), cerebrospinal fluid (CSF) (2, 3.1%), and urine (1, 1.5%).

### 3.2. Demographic and Clinical Profile

The cohort of 59 patients had a median age of 50 years (range: 11–86). The majority were male (75.0%). Chronic comorbidities were present in 11 patients (18.6% of 59 patients). Sixteen patients (27.1%) had a documented prior history of TB.

Clinical data were available for all 59 patients. All patients presented with symptoms suggestive of active TB (persistent cough, fever, dyspnea, weight loss, and fatigue). No asymptomatic patient was documented.

Pediatric cases: All four children with trace-positive results were clinically diagnosed with active TB and initiated treatment. Three of the four children (75%) had positive cultures, while the one culture-negative case had strong supporting evidence (known TB contact and cavitary lesion on CT scan).

An overview of the demographic and baseline clinical characteristics for the entire cohort is provided in Table 2.

### 3.3. Microbiological Correlation and Diagnostic Yield

The correlation between the 65 trace-positive samples and conventional bacteriological methods is detailed in Table 3. A total of 29 samples (44.61%) were confirmed positive for *M. tuberculosis* by culture. Only 5 samples (7.7%) were positive on smear microscopy, all of which were also culture-positive. Consequently, 24 of the 29 culture-positive samples (82.75%) were smear-negative, emphasizing the detection of paucibacillary cases by Xpert Ultra.

### 3.4. Results of Imaging Studies

Radiologic imaging (Chest X-ray or CT) was available for 56 of the 59 patients (94.91%). Among these, 47 patients (79.6%) had findings highly suggestive of active TB (e.g., upper lobe infiltrates, cavitary lesions, miliary patterns, or large pleural effusions), while 9 patients (15.2%) showed sequelae from prior TB. One patient with urinary TB had suggestive abdominopelvic imaging.

### 3.5. Final Diagnosis and Treatment Decisions

Based on the composite reference standard, 47 of the 59 patients (79.6%) were ultimately diagnosed with active TB (new case or relapse), while active TB was ruled out in 12 (20.4%) cases, 11 of whom had a history of TB within the preceding five years, suggesting residual DNA detection rather than active disease (Appendix A).

Of the 47 patients with active TB diagnosis, only 29 (61.7%) had a positive culture. Crucially, 18 patients (38.3%) were diagnosed and treated for active TB despite having a negative or contaminated culture. The decision to treat these cases was guided by the combination of the trace-positive result, highly suggestive symptomatology, and severe radiological findings.

Treatment decisions aligned with the diagnosis: 44 patients (74.5% of the 59) initiated anti-TB treatment. The discrepancy between the 47 diagnosed patients and the 44 treated patients is due to three individuals (two psychiatric patients and one homeless person) who were lost to follow-up or for whom treatment initiation could not be confirmed in the available registry data. Treatment was not initiated in 12 patients—11 with a documented history of tuberculosis within the previous five years, showing stable post-treatment sequelae on imaging, and one patient in whom active TB was ultimately ruled out despite an initial trace result (Figure 1).

## 4. Discussion

The present study assessed the clinical and diagnostic significance of Xpert MTB/RIF Ultra “trace” results in a real-world, high-incidence TB setting in Romania. Our findings provide necessary real-world evidence from a European high-burden setting, a context underrepresented in large international trials. Trace results accounted for 10.9% of all positive Xpert Ultra calls, consistent with reports from large multicenter analyses [5,35].

### 4.1. Diagnostic Yield and Correlation with Culture

The most compelling finding is that a trace call is a strong indicator of potential active disease in high-burden environments. Based on the CRS, 47 patients (79.7%) were ultimately diagnosed with active TB. Anti-TB treatment was initiated in 44 patients (74.6% of the cohort).

Of the 65 trace-positive samples, 44.6% (29/65) were culture-confirmed, closely mirroring rates reported in multicenter evaluations (34–57%) [6]. Only 7.7% (5/65) were smear-positive. This emphasizes the molecular assay’s capacity to detect infection where bacillary density is below the threshold of microscopy. Furthermore, 82.7% (24/29) of culture-positive trace samples were smear-negative, underscoring Xpert Ultra’s critical role in diagnosing paucibacillary TB.

Importantly, 18 patients (38.3% of all active TB cases) were diagnosed and treated despite negative or contaminated cultures. This demonstrates Xpert Ultra’s practical impact in enabling early diagnosis and reducing delays [25,28,36]. Clinical improvement following therapy further suggests most trace-positive results represent active infection [23].

Relying solely on culture provided a positive predictive value (PPV) of 49.2% (29/59). Against the CRS, the PPV increased to 79.7% (47/59). This significant improvement supports using CRS as a more realistic diagnostic framework for highly sensitive molecular assays in high-incidence settings [27,28].

According to WHO 2025 guidance, a trace-positive Xpert Ultra result may be considered bacteriological confirmation of TB when compatible clinical or radiological evidence is present, especially in high-risk groups such as children, HIV-positive patients, and those with extrapulmonary disease [19,28]. Conversely, in adults with previous TB or fibrotic sequelae, the probability of false positivity is higher due to detection of persistent non-viable bacilli [16]. In such cases, confirmatory culture or repeat testing with an alternative specimen type is recommended [5]. These findings confirm the added diagnostic value of Xpert Ultra trace results in paucibacillary and culture-negative TB and emphasize their role as a bridge between molecular detection and early clinical decision-making.

### 4.2. Clinical and Radiological Correlation

The decision to treat culture-negative, trace-positive patients was strongly supported by clinical and imaging findings. All patients were symptomatic at the time of testing. Among patients with available imaging (*n* = 56), 47 (83.9%) displayed features highly suggestive of active TB (e.g., cavitary lesions, miliary dissemination) [37,38].

Imaging evidence was the strongest determinant for treatment initiation in culture-negative cases. This integrated approach not only increases diagnostic yield but also prevents underdiagnosis of paucibacillary forms [5]. The strong concordance between trace positivity and compatible clinical findings confirms its predictive reliability in high-burden contexts [10].

### 4.3. Interpretation in Specific Patient Groups

The diagnostic weight of the trace call varies by patient subgroup:•Pediatric and EPTB: Findings reinforce high clinical significance in these groups. All four children with trace results were diagnosed and treated. Trace detection in EP samples (18.5% of all trace-positive specimens) was clinically meaningful, with pleural fluid showing 100% culture positivity. This supports the WHO recommendation to regard trace as bacteriological confirmation in children and EPTB, where conventional methods have limited sensitivity [5,18,22,27].•Patients with Prior Tuberculosis: Interpretation is more complex. Twelve patients with trace results were ultimately classified as non-active TB, with 11 having a documented prior history. This suggests trace may reflect residual DNA in this group [16,25]. WHO guidance advises that re-treatment should rely on clear clinical and radiological evidence of new activity [5]. Our data support this: patients with trace and prior TB required corroborating evidence, while trace positivity was almost always clinically significant in new TB suspects.

### 4.4. The Utility of a Composite Reference Standard

This study supports the adoption of a Composite Reference Standard (CRS) that integrates clinical, imaging, and microbiological criteria [27,28]. Relying solely on culture is fundamentally inadequate for highly sensitive molecular assays like Xpert Ultra, especially in paucibacillary disease. The CRS provides a pragmatic framework, allowing timely diagnostic and therapeutic decisions that better reflect real-world practice. Using culture as the sole reference, trace results had a positive predictive value (PPV) of 49.2% (29/59; 95% CI 36.8–61.6). When evaluated against the CRS, the PPV significantly increased to 79.7% (47/59; 95% CI 67.7–88.9).

A uniform diagnostic work-up ensured verification bias was unlikely. However, we acknowledged the potential for partial incorporation bias, as clinicians were aware of the Xpert Ultra result during decision-making [5,32]. To mitigate this, all cases were reclassified independently using predefined CRS criteria. The subgroup analysis provided critical nuance. While CRS showed high accuracy in Pediatric and Extrapulmonary cases (100% PPV), the most pronounced differences between culture-based and CRS-based classifications were observed in the pulmonary and previously treated subgroups. For patients with Prior TB (*n* = 22), the PPV rose from 18.2% (culture) to 50.0% (CRS).

This pattern suggests the improved yield under the CRS may reflect clinical over-classification of trace results in patients with fibrotic sequelae, highlighting the inherent trade-off in diagnostic decision-making. This underscores the need for cautious interpretation of trace results in adults with prior TB or residual lesions. WHO guidance similarly advises confirmatory testing before reinitiating therapy in these patients [5]. Overall, the CRS approach provides a balanced framework in high-incidence environments, integrating clinical and imaging data to enhance diagnostic sensitivity.

### 4.5. Interpretation Framework in Practice

Our data support a structured interpretive framework for Xpert Ultra trace results that integrates molecular findings with clinical and radiological context [4]. This approach ensures consistent diagnostic decisions and aligns with WHO recommendations for use in high-burden and high-risk populations.

The proposed tiered framework guides clinical judgment:•Trace + Active symptoms + Suggestive imaging + No prior TB → Strong justification for immediate treatment.•Trace + Immunocompromised host (e.g., HIV) + Compatible findings → Consider empirical treatment even if culture-negative.•Trace + Prior TB with stable sequelae and no new symptoms → Close monitoring and repeat testing before re-treatment.

This framework mirrors the evidence-based approach recommended by WHO and validated by recent clinical studies [23,27,37]. Operationally, adopting this model facilitates earlier treatment initiation, more accurate case classification, and better resource allocation in high-burden environments. The strong concordance between trace positivity and CRS-defined TB in our cohort reinforces the suitability of Xpert Ultra as a front-line diagnostic tool [4,5,33].

Despite clear WHO guidance, overreliance on automated molecular assays can lead clinicians to accept results uncritically. A balanced approach that integrates molecular findings with clinical and radiological context remains essential to ensure diagnostic accuracy and avoid both over- and under-diagnosis [5].

### 4.6. Limitations

Our study has several limitations inherent to its retrospective, single-center design. Relying on the CRS means the final diagnosis in culture-negative cases lacks strict bacteriological proof. Potential partial incorporation bias exists, as clinicians were aware of the Xpert Ultra result during the final diagnostic decision. Missing HIV status for 13 patients (≈22.0%) limits the full contextualization in this critical subgroup, where WHO guidance for “trace” is particularly strong. The inability to confirm treatment initiation for three individuals also limits the completeness of the real-world success rate.

Despite these issues, the study offers compelling clinical implications. By validating the high PPV of trace-positive results within a CRS framework, our findings support an integrated, context-sensitive diagnostic model. This approach moves beyond rigid reliance on culture, facilitates earlier diagnosis, and aligns clinical practice with the evolving goals of modern TB care and control.

## 5. Conclusions

In this high-TB-burden setting, Xpert MTB/RIF Ultra “trace” results hold substantial diagnostic and clinical significance, frequently reflecting true active, paucibacillary disease. When interpreted through a Composite Reference Standard, trace positivity is associated with active TB in nearly four-fifths of cases (79.7%). This confirms Xpert Ultra’s pivotal role in bridging the diagnostic gap in smear- and culture-negative TB. The findings validate the use of trace results as reliable evidence in clinical practice, emphasizing that an integrated, patient-centered diagnostic framework is necessary to maximize the diagnostic potential of this highly sensitive molecular assay and ensure timely patient management.

## Figures and Tables

**Figure 1 diagnostics-15-02860-f001:**
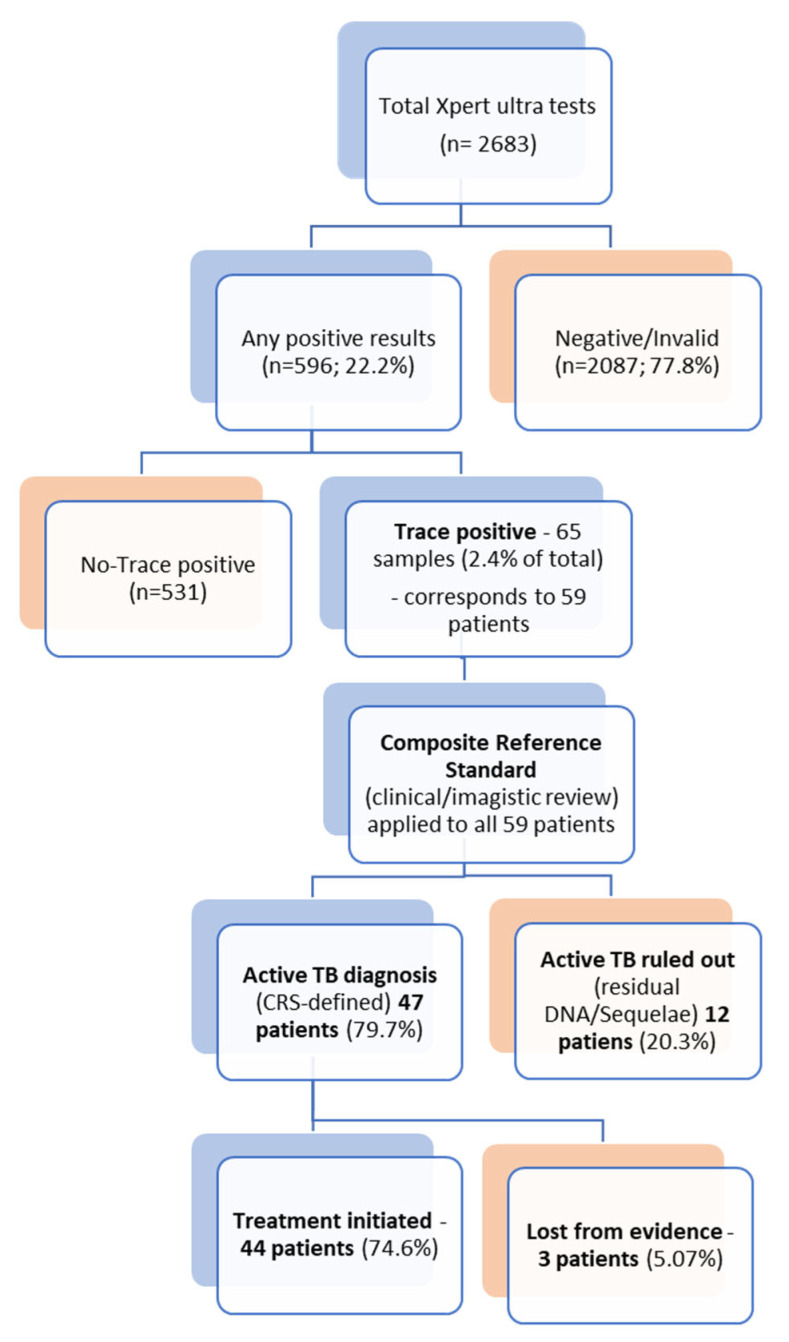
Diagnostic Pathway for Xpert Ultra Trace Results and Final Clinical Outcome.

**Table 1 diagnostics-15-02860-t001:** Distribution of trace-positive samples by specimen type.

Specimen Type	Number of Samples (*n* = 65)	Percentage (%)
Spontaneous Sputum	30	46.2%
Induced Sputum	23	35.4%
Pleural Fluid	6	9.2%
Bronchial Aspirate	3	4.6%
Cerebrospinal Fluid (CSF)	2	3.1%
Urine	1	1.5%
Total	65	100.0%

**Table 2 diagnostics-15-02860-t002:** Overview of the demographic.

Characteristic		*n* (%)
Total Patients		59 (100%)
Mean Age (years)		49.1
Sex	Male	44 (74.6%)
	Female	15 (25.4%)
Chronic Comorbidities		11 (18.6%)
Previous TB History		16 (27.1%)

**Table 3 diagnostics-15-02860-t003:** Correlation between Culture Result and Final Therapeutic Decision (N = 59 Patients).

Therapeutic Decision (Anti-TB Treatment)	Culture Positive	Culture Negative/Contaminated	Total Patients *n* (%)
Treated	26 (5 Smear Positive)	18 *	44 (74.6%)
Not Treated	3	12	15 (25.4%)
Total (*n*)	29	30	59 (100.0%)

* Note: Patients treated with a Culture Negative result (*n* = 18) were diagnosed as active TB based on the CRS (trace-positive + suggestive symptoms + suggestive imaging).

## Data Availability

The original contributions presented in this study are included in the article/Appendix A. Further inquiries can be directed to the corresponding author.

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
