# Peer review of "Clinical Relevance of Trace-Positive Results in Xpert MTB/RIF Ultra for Tuberculosis Diagnosis in a High-Burden Setting: A Retrospective Cohort Study"

_diagnostics, 2025, doi:10.3390/diagnostics15222860_

Round 1

Reviewer 1 Report

Comments and Suggestions for Authors

I could not get the implications from the manuscript. All the context presented have been reported in previous literatures. In addition, the data is not enough to support the perspective hold by the authors. Another independent validation is needed to demonstrate the authors’ concept.

Author Response

We thank the reviewer for their careful assessment of our manuscript and for raising critical points regarding the novelty and data robustness of our work. We understand the concern that the context of "trace" results has been reported in the literature, and we want to take this opportunity to clarify the specific and necessary contribution of our study. The unique implication of our findings lies in providing essential real-world evidence from a high-incidence European Union setting, specifically Romania, which carries the highest TB burden in the EU/EEA. This epidemiological context is distinct and requires tailored data, as it differs significantly from studies conducted in high-HIV prevalence regions or very low-incidence settings. Our work addresses this gap by showing how the highly sensitive Xpert Ultra assay performs and, more importantly, how the "trace" result influences clinical decision-making in our local practice.

Furthermore, our study is particularly timely and relevant given the recent evolution of international diagnostic guidance. The clinical interpretation and weight assigned to Xpert Ultra "trace" results have undergone a significant shift in the latest WHO consolidated guidelines (2025), moving towards a more aggressive and integrated approach compared to prior recommendations (e.g., 2021). Our data provide strong local validation for this updated perspective, as we found that when the trace result was interpreted using a Composite Reference Standard (CRS) that integrates clinical and radiological findings, the positive predictive value rose sharply to 79.7%. This directly supports the new WHO recommendation that in high-burden environments and high-risk patients, a trace call often signifies true, active, paucibacillary disease, enabling crucial, early treatment initiation.

Regarding the perceived insufficiency of our data, we respectfully emphasize that our cohort of 59 patients represents a robust and comprehensive collection of the rarest diagnostic subset: the "trace" result itself. This finding accounted for only 2.4% of all Ultra tests performed over six years in our center. Focusing exclusively on this low-frequency, yet clinically challenging, group over a prolonged period provides valuable insight into real-world diagnostic complexity. We used the CRS precisely to overcome the inherent limitations of culture as a gold standard in paucibacillary cases, and the sharp increase in our PPV when applying this integrated framework confirms the clinical validity of our diagnostic and therapeutic decisions. We believe that this contextual validation, demonstrating how the trace result reliably guides treatment in an under-reported European setting, strengthens the scientific literature and supports the adoption of modern, integrated diagnostic algorithms.

To ensure these key aspects are fully appreciated by the readership, we have incorporated revisions into the Introduction and Discussion to more clearly articulate the novelty of the European context and the timely validation of the evolving WHO recommendations.

We sincerely hope that these clarifications and the revisions made to the manuscript satisfy the reviewer's concerns and demonstrate the substantial contribution of our study to the field.

Sincerely,

The Authors.

Reviewer 2 Report

Comments and Suggestions for Authors

Dear Authors,

I am very glad to review your manuscript entitled “Clinical Relevance of Trace-Positive Results in Xpert MTB/RIF Ultra for Tuberculosis Diagnosis in a High-Burden Setting.” The topic is clinically relevant and well-aligned with WHO’s current focus on optimizing molecular TB diagnostics. The addition of the supplementary CRS dataset enhances the transparency and credibility of your findings by allowing a case-level assessment of data completeness and classification.

"Discussion section" remains excessively long and repetitive, it dilutes the manuscript’s impact. It reiterates several points from the Introduction and Results (e.g., interpretation of “trace” results, WHO guidance, and the diagnostic role of Xpert Ultra). I strongly recommend a major revision focused on condensing and restructuring the Discussion around key themes such as: Diagnostic yield and correlation with culture; Clinical and radiological concordance; Interpretation in special populations (children, EPTB, previously treated TB); Limitations and clinical implications.

My Specific recommendations are also: The supplementary CRS table is helpful; consider summarizing it visually (e.g., a concise comparative table or figure showing CRS-based classification vs. culture results, with treatment initiation status). Include a simple flow diagram (total tests → positives → trace → CRS-defined active TB → treatment started).Simplify Tables 3 and 4 by grouping treated vs. untreated cases to emphasize clinical decision pathways.Use consistent terminology (Xpert Ultra, TB, EPTB). Revise the Conclusion to highlight key take-home messages rather than restating results.

Overall, this is an important and potentially publishable study once the Discussion is shortened and the results presented in a more integrated and visual format.

With best regards,

Comments on the Quality of English Language

Many parts of the Discussion use long, multi-clause sentences. Shorter, more direct sentences will make the paper clearer and easier to follow (e.g., Replace “This finding, which is consistent with WHO guidance and prior studies, highlights...” with “This finding aligns with WHO guidance and previous studies.”) Please, use "past tense" for methods and results, "present tense" for interpretation and conclusions.

Author Response

We thank the reviewer for their thorough assessment, positive feedback on the clinical relevance of our topic, and constructive guidance. We have performed a major revision of the manuscript, specifically addressing every structural, visual, and language-related point raised in your review.

  • We completely restructured the Discussion section to follow the suggestion, and we have significantly condensed this section, structuring it according to the suggestions.
  • We replaced the previous Table 4 with a dedicated figure (Figure 1: Diagnostic Pathway for Xpert Ultra Trace Results and Final Clinical Outcome). This simple flow diagram shows the progression from Total Tests → Positives → Trace Results → CRS-defined Classification → Treatment Started, providing an integrated visual summary of the patient cohort and diagnostic decisions.
  • The previous Table 4 was merged into a revised Table 3 (Correlation between Culture Result and Final Therapeutic Decision). This new table directly compares the number of treated versus non-treated patients against their final culture status (Positive vs. Negative/Contaminated).
  • We reviewed the entire manuscript and ensured consistent terminology is used, especially abbreviations.

 We believe these substantial revisions have significantly improved the structure, clarity, and impact of the manuscript. We appreciate the reviewer's valuable input.

Sincerely,

The Authors.

Round 2

Reviewer 1 Report

Comments and Suggestions for Authors

It is okay for me

Author Response

We sincerely thank the Reviewer for their time and positive evaluation of our manuscript.

Reviewer 2 Report

Comments and Suggestions for Authors

Dear Authors;

Thank you for the corrections. I have also included some additional revisions in the text. 

Best regards

Comments on the Quality of English Language

Many parts of the Discussion use long, multi-clause sentences. Shorter, more direct sentences will make the paper clearer and easier to follow (e.g., Replace “This finding, which is consistent with WHO guidance and prior studies, highlights...” with “This finding aligns with WHO guidance and previous studies.”) Please, use "past tense" for methods and results, "present tense" for interpretation and conclusions.

Author Response

We sincerely thank the Reviewer for the valuable time dedicated to the review process and for the detailed examination of the manuscript. We have carefully reviewed and accepted all additional textual revisions and minor corrections the Reviewer included directly in the text. These suggestions have further improved the clarity and readability of the manuscript.

Best regards